# Genetic Diversity and Population Structure of *Mycobacterium bovis* at the Human-Animal-Ecosystem Interface in France: “A One Health Approach”

**DOI:** 10.3390/pathogens12040548

**Published:** 2023-04-01

**Authors:** Anaïs Appegren, Maria Laura Boschiroli, Krystel De Cruz, Lorraine Michelet, Geneviève Héry-Arnaud, Marie Kempf, Philippe Lanotte, Pascale Bemer, Olivia Peuchant, Martine Pestel-Caron, Soumaya Skalli, Lucien Brasme, Christian Martin, Cecilia Enault, Anne Carricajo, Hélène Guet-Revillet, Michaël Ponsoda, Véronique Jacomo, Anne Bourgoin, Sabine Trombert-Paolantoni, Christian Carrière, Chloé Dupont, Guilhem Conquet, Lokman Galal, Anne-Laure Banuls, Sylvain Godreuil

**Affiliations:** 1Laboratory of Bacteriology, CHU Montpellier, 34000 Montpellier, France; 2ANSES Laboratory for Animal Health, Tuberculosis National Reference Laboratory, University Paris-Est, 94000 Maisons-Alfort, France; 3Laboratory of Bacteriology, CHU Brest, 29000 Brest, France; 4Laboratory of Bacteriology, CHU Angers, 49000 Angers, France; 5Laboratory of Bacteriology, CHU Tours, 37000 Tours, France; 6Laboratory of Bacteriology, CHU Nantes, 44000 Nantes, France; 7Laboratory of Bacteriology, CHU Bordeaux, 33000 Bordeaux, France; 8Laboratory of Bacteriology, CHU Rouen, 76000 Rouen, France; 9Laboratory of Bacteriology, CHU Reims, 51000 Reims, France; 10Laboratory of Bacteriology, CHU Limoges, 87000 Limoges, France; 11Laboratory of Bacteriology, CHU Nîmes, 30000 Nîmes, France; 12Laboratory of Bacteriology, CHU Saint-Etienne, 42000 Saint-Etienne, France; 13Laboratory of Bacteriology, CHU Toulouse, 31000 Toulouse, France; 14Laboratoires Biomnis Eurofins, 69000 Lyon, France; 15Laboratory of Bacteriology, CHU Poitiers, 86000 Poitiers, France; 16Cerba Europeanlab, 95000 Saint-Ouen l’Aumone, France; 17UMR, MIVEGEC, IRD, CNRS, Université de Montpellier, 34000 Montpellier, France

**Keywords:** bovine tuberculosis, population genetics, zoonosis, One Health, France

## Abstract

*Mycobacterium bovis* infects cattle and wildlife, and also causes a small proportion of tuberculosis cases in humans. In most European countries, *M. bovis* infections in cattle have been drastically reduced, but not eradicated. Here, to determine the *M. bovis* circulation within and between the human, cattle, and wildlife compartments, we characterized by spoligotyping and mycobacterial interspersed repetitive unit-variable number tandem repeat (MIRU-VNTR) typing the genetic diversity of *M. bovis* isolates collected from humans, cattle, and wildlife in France from 2000 to 2010. We also assessed their genetic structure within and among the different host groups, and across time and space. The *M. bovis* genetic structure and its spatiotemporal variations showed different dynamics in the human and animal compartments. Most genotypes detected in human isolates were absent in cattle and wildlife isolates, possibly because in patients, *M. bovis* infection was contracted abroad or was the reactivation of an old lesion. Therefore, they did not match the genetic pool present in France during the study period. However, some human-cattle exchanges occurred because some genotypes were common to both compartments. This study provides new elements for understanding *M. bovis* epidemiology in France, and calls for increased efforts to control this pathogen worldwide.

## 1. Introduction

Most emerging and reemerging human diseases are caused by zoonotic agents [1]. Human tuberculosis (TB) is caused mainly by *Mycobacterium tuberculosis*. However, there is a lot of evidence that suggests that the contribution of *Mycobacterium bovis*, the agent of bovine TB, might be underestimated in humans [2]. The World Health Organization (WHO) has recently recognized zoonotic TB due to *M. bovis* as a neglected disease, and has integrated it into its global TB control program called "WHO end TB strategy". Understanding and controlling these diseases require integrated approaches that take into account pathogens, humans, wildlife, and domestic animals [2]. This "One Health" strategy is recommended by the WHO to promote a common human and veterinary approach against zoonoses, including zoonotic TB [3,4,5]. *M. bovis* infection epidemiology is particularly complex because of its wide host range, including humans and many wild (e.g., badgers, possums) and domestic animals (e.g., cattle, goat, sheep) [6,7,8]. *M. bovis* is mostly transmitted to humans through the consumption of unpasteurized dairy products, carcass handling, and by the airborne route when in close contact with infected animals [9,10]. In European countries, where milk pasteurization and test-and-cull protocols in cattle have been implemented in the last five decades, *M. bovis* causes 1 to 3% of human TB cases [4]. This rate might be much higher in developing countries, especially in regions where *M. bovis* infections are enzoonotic and where HIV, which may act as an epidemiological catalyst, is endemic [3,4,5]. It is assumed that in industrialized countries, most human cases concern elderly people (reactivation of an ancient infection) and people who were infected in a foreign country [11]. However, a genetic fingerprinting study in Spain, England, and Ireland challenged this assumption by showing that the genotype of 47% of human isolates was identical or similar to that of bovine isolates [9,10,11,12]. As human isolates were mostly from elderly people but not from migrants, the authors suggested that in industrialized countries, recent infections of *M. bovis* could be more frequent than initially thought [9,12]. 

In France, the rate of human TB caused by *M. bovis* was 0.09% in 2015 (n = 198 microbiologically documented TB cases between 2011 and 2016) and concerned mainly people born in an endemic country [13,14]. In this French study, no recent animal-to-human transmission was identified [13,14]. Human-to-human transmission can rarely occur, as suggested by molecular epidemiology studies [9,10,15,16,17]. 

*M. bovis* infections among animals are due to aerosol transmission [6] and also environmental transmission because *M. bovis* can remain active for days or even month in its host’s feces [18]. Transmission can occur among and within the cattle and wildlife compartments [8,19]. Moreover, cases of domestic animals contaminated with *M. Tuberculosis* by human (i.e., an anthropozoonosis) excreta have been reported [20]. Wildlife reservoirs of *M. bovis*, such as badgers in the UK, cervids in the USA [21], and possums in New Zealand [1], are a major challenge for disease control. Therefore, this multi-host pathogen has important implications for human health, the cattle sector economy, and wildlife conservation. On the basis of recent findings on *M. bovis* genetics, Allen et al. suggested that some strains could be more adapted for some host types [22]. Indeed, most *M. bovis* strains harbor a mutation in a virulence operon (phoPR) that reduces their virulence in humans, but not in non-human hosts [23]. However, in some *M. bovis* strains involved in human-to-human transmission events, a second mutation was observed that restores its virulence potential in humans, therefore “re-adapting” these strains to the human host [23]. Moreover, a comparative genomic study of different *M. bovis* strains and one *Mycobacterium caprae* strain from cattle and wild boar suggested that specific alterations in ESX loci, which are involved in mycobacterial viability or virulence, may be implicated in host tropism because the same isolate could behave differently in different hosts [24]. These observations suggest that different *M. bovis* strains could be more adapted to different host types. Moreover, the fact that drastic control measures in cattle in most industrialized countries for the last five decades have not eradicated *M. bovis* suggests that its transmission mechanisms have not been fully elucidated, and that studies are necessary to understand the transmission among the different host compartments.

The choice of the optimal molecular markers in accordance with the scope of the study also depends on the space and time scales in which the data were collected or explored. In fact, the speed of evolution (molecular clock) of a given marker conditions its power of resolution. Nevertheless, the resolution power of each marker is a function of the organism and the species under study. For *M. bovis*, from a resolution power point of view, MIRU-VNTR appeared very appropriate for short-term epidemiology studies, whereas spoligotyping is more suitable for long-term epidemiology studies. Nevertheless, several studies showed clearly that using multiple methods for molecular epidemiology is necessary [25,26,27,28]. Several authors recommend spoligotyping associated with MIRU-VNTR for molecular epidemiology studies [25,26,27,28]. Several authors recommend spoligotyping associated with MIRU-VNTR for molecular epidemiology studies [25,26,27,28]. 

In France, *M. bovis* genetic diversity has been extensively described in cattle and wild animals [8]. Specifically, the genetic diversity of cattle-isolated strains has been decreasing since the 2000s, probably as a consequence of control measures and/or changes in agricultural practices, and most genotypes seem to be associated with specific spatial locations [8]. Moreover, common genetic fingerprints were identified in cattle and wildlife isolates. Deer, wild boar, and badgers are probably involved in *M. bovis* exchanges between cattle and wildlife in several French regions, and may act as local *M. bovis* maintenance hosts [29]. However, to date, no study has assessed and compared the genetic diversity of *M. bovis* isolates obtained from infected humans and animals in France. Therefore, here, we compared the *M. bovis* genetic diversity in isolates collected from infected humans, cattle, and wildlife by spoligotyping and mycobacterial interspersed repetitive unit-variable number tandem repeat (MIRU-VNTR) typing, two standard genetic fingerprinting and population genetic methods. We assessed the *M. bovis* genetic structure within and among the different host groups, and across time and space. We also investigated whether traveling abroad may play an important role in *M. bovis* epidemiology in the human compartment, and whether *M. bovis* isolates involved in pulmonary and extrapulmonary TB forms in humans are genetically different. 

## 2. Material and Methods

### 2.1. Ethics Statement

All *M. bovis* isolates from animals (cattle and wildlife) were obtained from the French National Reference Laboratory for Tuberculosis, Animal Health Laboratory, ANSES. This laboratory is in charge of the surveillance of bovine tuberculosis in farm and wild animals in France, and is also in charge of the phenotypic and genotypic characterizations of *M. bovis* isolates in agreement with national regulations. As the study did not involve invasive procedures on live animals, no ethical approval was necessary. All human *M. bovis* isolates were collected during routine diagnostic microbiology procedures. As clinical bacterial isolates are not considered to be human biological samples (unlike serum and tissue biopsies), no informed consent was required for their use in research. Lastly, all retrospective epidemiological data associated with bacterial isolates were anonymized and therefore not traceable. Therefore, for this study, no ethical approval was necessary. 

### 2.2. Sample and Information Collection

In this study, 1451 non-redundant *M. bovis* (1446, 99.65%) and *M. caprae* (5, 0.35%, exclusively in humans) isolates were used. They were collected in France from 2000 to 2010, and included 214 human isolates, 1106 bovine isolates, and 131 isolates from wild animals (66 boars, 46 badgers, 19 deer). All information on these isolates is summarized in Appendix A (human samples) and Appendix A (animal samples) in the Appendix A. Sample size by host species and the distribution of genetic patterns (spoligotypes, MIRU-VNTRs and spoligotypes + MIRU-VNTRs) by year and region are presented in Figure 1 and Figure 2.

Human dataset: Human TB cases caused by *M. bovis* in France were identified using a standardized questionnaire sent to laboratories of the French national microbiological network. This network gathers the microbiological laboratories of the 29 French university hospitals, the Pasteur Institute Mycobacteria Reference Laboratory (Paris), and two reference private laboratories (Biombis and Cerba) that perform most mycobacterial cultures for non-academic hospitals. The reporting quality was checked by comparing with data from the French National Reference Center for the Surveillance of Mycobacterial Diseases and Drug Resistance and the French Institute for Health Monitoring. All *M. bovis* isolates collected from patients with zoonotic TB in metropolitan France (but not for Corsica and Basse-Normandie from which no information could be obtained) from 2000 to 2010 were included in the study. Data were spatially aggregated in 19 regions (i.e., the administrative division of France before 2016, Figure 2). For almost each patient, information about age, sex, country of birth and travels to foreign countries, clinical data (pulmonary or extrapulmonary form of TB), and risk factors (childhood contact, consumption of raw milk, infected family member, occupational risk for breeders, butchers, and slaughterhouse workers) were collected.

Cattle and wildlife dataset: All *M. bovis* isolates from animals (cattle and wildlife) collected in metropolitan France from 2000 to 2010 were included in the study. Data on these animals were exhaustively described in Hauer et al., 2015 [8]. Isolates are routinely collected by the French National Reference Laboratory (NRL) for Bovine Tuberculosis on behalf of the French Agriculture Ministry in the framework of bovine tuberculosis eradication and surveillance programs. In infected cattle, *M. bovis* is detected by bacterial culture after slaughtering for diagnostic purposes following a previous positive skin test or interferon-γ assay, or after detection of macroscopic TB-like lesions during the routine post-mortem meat inspection. Wildlife surveillance programs have been implemented in French regions where bovine TB is important and in the framework of the Sylvatub program launched in 2011 [8]. To standardize the human and animal datasets, data on animal isolates were aggregated at the same administrative level. 

### 2.3. Mycobacterium Culture and Species Identification

Human respiratory and extra-respiratory specimens were digested and decontaminated using the sodium dodecyl sulfate–NaOH method and then centrifuged. Samples from all patients were stained to detect acid-fast bacilli, and cultured on solid and liquid media (BACTEC MGIT 960 System; Becton Dickinson Diagnostic Systems, Franklin Lakes, NJ, USA, https://www.bd.com, accessed on 21 March 2023). A commercial kit (GenoType^®^ MTB assay; Hain Lifescience; Bruker-Hain Diagnositics, Nehren, Germany, https://www.hain-lifescience.de, accessed on 1 March 2022) was used to identify isolates as Mycobacterium tuberculosis bacterial complex (MTBC) species, particularly *M. bovis* and *M. caprae*. Drug susceptibility testing was performed with the MGIT 960 system.

Samples from cattle and wildlife were cultured using the protocol established by the French NRL (NF U 47–104) for *M. bovis* isolation. Following decontamination, the supernatant was seeded on Löwenstein–Jensen and on Coletsos culture media, incubated at 37 °C ± 3 °C for 3 months, and examined every 2 weeks. MTBC species were identified by DNA amplification of colonies, as described by Hénault et al. (2006) [30], and *M. bovis* was identified by spoligotyping (see below). 

### 2.4. Genotyping Methods

Human, cattle, and wildlife isolates were genotyped by spoligotyping and MIRU-VNTR analysis. For cattle, at least 1 isolate/herd/outbreak was genotyped using both methods. If there was more than one spoligotype profile in the same herd in a given year, two isolates for each spoligotype profile were genotyped also by MIRU-VNTR. Spoligotyping was performed, as described by Kamerbeek et al. [31], using spoligotyping membranes that include the 43 spacer sequences present in the DR locus (Isogen Bioscience BV, Maarssen, The Netherlands), or by hybridization onto fluorescent microbeads coupled to spacer sequences followed by laser detection (Luminex), as described by Zhang et al. [32]. The presence/absence of the 43 spacers contained in the DR locus was represented by a binary code of 43 entries. Spoligotypes were named according to the international convention, using the www.mbovis.org (accessed on 21 March 2023) database. MIRU-VNTR profile identification [33,34] was performed by Genoscreen, Lille, France, by PCR amplification to target eight mycobacterial loci (ETR A, B, C, D, QUB11a, QUB11b, QUB 26, VNTR 3232).

### 2.5. Genetic Diversity and Population Structure Analyses

#### 2.5.1. Spoligotype Frequencies and Comparisons of Complete Genotypes (Spoligotypes + MIRU-VNTRs) 

The frequencies of the different spoligotypes were determined in the whole dataset and for each host. The ratio between the number of spoligotypes and the number of genotyped isolates was calculated as a genetic diversity index for a given host. Spoligotypes with a frequency > 5% were considered as the “main” spoligotypes in that host. For all shared spoligotypes, the presence of shared MIRU-VNTR profiles was investigated to highlight possible human-cattle and human-wildlife transmissions. 

The presence of isolates belonging to the F4 family in the human dataset was checked. This family is defined by the absence of spacer 33 in the DR region and the presence of truncated repeats in the MIRU-VNTR locus 4052 (QUB26), and is frequent in cattle in the south of France (Hauer et al., 2015) [8].

#### 2.5.2. MIRU-VNTR-Based Genetic Differentiation Analyses

The effects of host type (human, cattle, or wildlife), space (sampling region), and time (sampling year) on the *M. bovis* genetic structure (excluding the five *M. caprae* isolates) were tested using the MIRU-VNTR data. First, the whole dataset (human, cattle, and wildlife isolates) was considered, and then it was divided into three groups according to the host. As some human cases could be due to the reactivation of an old infection in elderly people (contracted before the implementation of control measures) or to infections contracted abroad, these analyses were performed again by focusing only on <50-year-old patients who were born in France and who had never travelled abroad. Moreover, the analysis was repeated by comparing all human, cattle, and wildlife isolates harboring the three most abundant spoligotypes. Then, the whole dataset was separated according to the sampling year (one group per year) and region, for the three hosts together and separately.

For each group, the mean genetic diversity (h) with its standard deviation was computed for the eight loci, and the population structure among these groups was determined using the FST value (an index of genetic differentiation that ranges from 0 = no differentiation to 1 = all samples are fixed for a different allele). These parameters were calculated using FSTAT v2.9.3.2 [35]. Genetic diversity differences among groups were tested using ANOVA and the Tukey post-hoc test, or the Welch’s *t*-test implemented in R v. 3.3.1 (R Core Team, 2016), Vienna, Austria. The standard error of the FST was computed by jackknifing over loci, and its significance was tested based on 10,000 randomizations using FSTAT v2.9.3.2. Pairwise FST between host compartments were also computed with their significant tests based on 60 permutations in FSTAT v2.9.3.2.; because of multiple comparisons, the significant threshold for this test was 0.017.

To quantify the relative impact of host species and space on the genetic structure, hierarchical analyses of molecular variance (AMOVA) were performed using the Excel-implemented program Genalex v. 6.5 (PE, 2012) by focusing on the seven French regions with N ≥ 9 human and cattle isolates and on the two French regions with sufficient cattle and wildlife isolates. The analysis with all three hosts could not be performed because only one region had a sufficiently high number of cases in all three hosts (Aquitaine). In this analysis, the host compartment was considered as nested in space and vice versa, and the significance of each level was tested based on 999 permutations. As both analyses gave similar results, only the results for the host compartment nested in the region are presented.

Finally, to obtain visual information on the *M. bovis* genetic structure, a discriminant analysis of principal components (DAPC) was performed using the adegenet 1.4-2 package (Jombart, 2008) in R v3.3.1 [36]. This analysis combines a principal component analysis (PCA; considering inter-individual variations) with a discriminant analysis (DA; considering inter-group variations). Based on cumulative variance and eigenvalues, 40 principal components were retained for the first step (PCA), and two discriminant functions for the second step (DA).

## 3. Results

### 3.1. Study Sample

This analysis concerned 1451 non-redundant *M. bovis*/*M. caprae* isolates collected in metropolitan France from 2000 to 2010: 214 human isolates (the human isolates of *M. bovis* number 215 and 216 were added to Appendix A, but were not included in the genetic analyses), 1106 bovine isolates, and 131 isolates from wild animals (66 boars, 46 badgers, 19 deer) (Appendix A).

### 3.2. Spoligotype Diversity among and within Host Species

#### 3.2.1. Spoligotype Diversity in Each Host Type

Spoligotyping led to the identification of 120 different spoligotype signatures (Appendix A). All animal isolates were identified as *M. bovis*, but the human dataset also included six *M. caprae* isolates (corresponding to the following orphan spoligotypes: SB0157, SB0415, SB0416, SB0853, SB0866, and SB1916). In samples isolated from patients (n = 214), 78 different spoligotypes (ratio = 0.36) were identified: 23 spoligotypes were detected in 157 isolates, whereas 55 spoligotypes were identified in one single isolate/each. Cattle samples (n = 1106) could be grouped into 68 spoligotypes (ratio = 0.06): 49 clusters comprised 1087 isolates, and 19 spoligotypes were represented by a single isolate. All wildlife samples (n = 131) presented four spoligotypes (ratio = 0.03). One human isolate included 19 spoligotypes not present in the http://www.mbovis.org (accessed on 21 March 2023) database, to which new codes (from SB2397 to SB2415) were attributed. 

When considering all host groups together, the most frequent spoligotypes were SB0120 (34.0%), SB0134 (16.3%), SB0825 (7.4%), and SB0121 (6.8%). In human samples, the main spoligotypes were SB0120 (22.0%), SB0121 (13.6%), SB0134 (10.3%), and SB0265 (5.6%). In cattle samples, the main spoligotypes were SB0120 (36.1%), SB0134 (13.0%), SB0825 (9.8%), SB0827 (6.1%), SB0121 (5.8%), and SB0821 (5.5%). In cattle, SB0265 was identified in only one sample (out of 1106) in an animal from Spain. In wildlife samples, the main spoligotypes were SB0134 (54.2%), SB0120 (36.6%), SB0821 (5.3%), and SB0121 (3.8%). Spoligotype frequencies were not balanced in the three wildlife species. In boars, strains clustered into four spoligotypes: SB0120 (13.6%), SB0121 (1.5%), SB0134 (74.2%), and SB0821 (10.6%). In badgers, SB0120 and SB0134 were detected in 82.6% and 17.4% of strains, respectively. In deer, the frequencies of SB0120, SB0121, and SB0134 were 5.3%, 21.1%, and 73.7%, respectively. The human dataset included 51 spoligotypes that were not present in the animal datasets, and the cattle dataset included 40 spoligotypes not present in the human and wildlife datasets (Figure 3). Human and cattle samples shared 24 spoligotypes, cattle and wildlife isolates shared one spoligotype, and the three host types had three spoligotypes (SB0120, SB0121, and SB0134) in common. 

#### 3.2.2. Spatiotemporal Variation in Spoligotype Diversity for Each Host Type

The total number of spoligotypes detected in human samples varied according to the collection year, possibly linked to yearly variations in the number of collected isolates (Figure 1a,b). Conversely, the number of spoligotypes remained rather stable in cattle samples, although the number of isolates strongly increased in 2010 (Figure 1c,d). In wild animals, the number of spoligotypes per year showed small variations (from 0 to 4 spoligotypes) that did not seem to be related to the number of collected isolates (Figure 1e,f).

Spoligotype diversity in human samples was highest in regions with the highest total number of isolates: Île de France (n = 48) and Rhône-Alpes (n = 40) (Figure 2(b1)). For cattle samples, spoligotype diversity was also high in regions with many isolates (e.g., Aquitaine: n = 377), but not always. For instance, the number of spoligotypes was higher in the Midi-Pyrénées region (n = 60 isolates) than in the Languedoc-Roussillon region (n = 167 isolates) (Figure 2(b2)). The spatial variation in the number of spoligotypes was very limited for wildlife samples, due to the low sample size and global diversity of isolates (Figure 2(b3)).

### 3.3. Diversity of MIRU-VNTR Profiles and the Combination of Spoligotypes and MIRU-VNTR Profiles

Spoligotypes and MIRU-VNTR profiles (hereafter, their combination is called “complete genotypes”) were obtained from 204 human isolates, 967 cattle isolates, and 100 wildlife isolates (n = 45 from badgers, n = 7 from deer, and n = 48 from boars). This combination of markers allowed for defining 497 different complete genotypes. 

In humans, all isolates were grouped into 194 MIRU-VNTR profiles and 194 complete genotypes. This means that each MIRU-VNTR profile was associated with a single spoligotype (ratio = 0.94). Most complete genotypes represented unique isolates; however, twelve genotypes were identified in two isolates, and one in three isolates (clusters: A, B, C, D, E, F, G, H, I, J, K, L, Appendix A). Three of these isolates harboring shared complete genotypes corresponded to documented cases of intra-human transmission (clusters A, B, C)15–17.

Cattle isolates were defined by 235 MIRU-VNTR profiles and 313 complete genotypes (ratio = 0.24 and 0.32, respectively). This indicated that several MIRU-VNTR profiles were associated with different spoligotypes. Each complete genotype was identified in one to 106 isolates. The most abundant genotypes were, as defined in Hauer et al. (2015) [8]: SB0120-5-3-5-3-9-4-5-6 (10.9%), SB120-5-5-4-3-11-4-5-6 (8.2%), and SB0134-6-4-5-3-6-4-3-6 (6.2%). 

The wildlife isolates with complete genotypes were characterized by 15 different MIRU-VNTR profiles and 15 complete genotypes (ratio = 0.15). Therefore, each MIRU-VNTR profile was associated with a single spoligotype. Each complete genotype was detected in one to 34 isolates. The most abundant genotypes in wild animals were: SB0134-6-4-5-3-6-4-3-6 (34%) and SB0120-5-5-4-3-11-4-5-6 (22%). SB0120-5-3-5-3-9-4-5-6, the most important profile in cattle, was detected in 12% of all wildlife isolates. The complete genotypes tended to be different in the three wildlife species, and SB0120-5-3-5-3-9-4-5-6 was the only genotype detected in the three host compartments. Clusters including cattle and wildlife isolates may represent possible cases of bovine-to-bovine and of cattle-to-wildlife transmission, as already described in Hauer et al. (2015) [8]. Particularly, exchanges between cattle and deer, wild boar and badgers were identified (Hauer et al., 2015) [8]. The present study identified three genotypes shared by human and cattle isolates [ SB0837-5-4-3-3-10-2-5-7 (1 human, Rhône-Alpes and 3 cattle, Pays-De-La-Loire); SB0265-5-4-3-3-11-2-4-6 (2 human, Languedoc-Roussillon/Morocco and 1 cattle, Franche-Comté); SB0816-5-5-5-3-8-2-5s-7 (1 human, Auvergne and 2 cattle, Poitou-Charentes)], suggesting possible cases of recent interspecific transmission. 

The most abundant complete genotypes in animals (i.e., SB0120-5-3-5-3-9-4-5-6, SB120-5-5-4-3-11-4-5-6, and SB0134-6-4-5-3-6-4-3-6) were absent in human isolates. Only three human isolates met the criteria that define the F4 family: the presence of SB0832-6-5-5-3-11-2-5s-8, SB0818-5-5-5-3-8-2-5s-7, and SB2413-7-5-5-3-11-2-5s-8. These three isolates were from two patients from Auvergne and one patient from Pays-de-la-Loire, two regions that are not in the south of France where F4 has been detected in cattle. 

### 3.4. Spatiotemporal Variation in the Number of MIRU-VNTR Profiles and Genotypes

As observed for the spoligotypes, the number of MIRU-VNTR profiles and complete genotypes in human samples varied in time according to the number of collected isolates (Figure 1a,b). Conversely, the same number of MIRU-VNTR profiles and complete genotypes was detected in a given year (Figure 1a,b). In cattle samples, the number of MIRU-VNTR profiles and complete genotypes globally decreased over time, and presented an important drop in 2001 and 2002 (Figure 1c,d). For each year, the number of complete genotypes was higher than the number of MIRU-VNTR profiles, because the same MIRU-VNTR profile could be associated with different spoligotypes (Figure 1c,d). In wildlife samples, the number of MIRU-VNTR profiles and complete genotypes tended to increase over time, probably due to an increase in sampling efforts, but remained relatively low (Figure 1e,f). Each year, the number of complete genotypes was identical to the number of MIRU-VNTR profiles. 

As observed for the spoligotypes, the number of MIRU-VNTR profiles and complete genotypes in the human dataset varied spatially according to the sample size, with the highest diversity in Île de France and Rhône-Alpes (Figure 2(b1)). For the cattle dataset, diversity was especially high in the south of France and in Bourgogne, regions with high sample sizes (Figure 2(b2)). The spatial variation in the genotype diversity was very limited for the wildlife dataset, but was highest in Bourgogne (n = 73 samples collected and n = 73 samples genotyped) (Figure 2(b3)).

### 3.5. Genetic Differentiation (MIRU-VNTR Data)

#### 3.5.1. Among Host Species

Analyses based on the MIRU-VNTR data were performed by including only isolates with complete genotypes, except for the six *M. caprae* isolates that were excluded from this step. The mean genetic diversity of *M. bovis* was 0.676 ± 0.167 in human samples, 0.570 ± 0.202 in cattle samples, and 0.428 ± 0.228 in wildlife isolates. Differences in genetic diversity were significant between human and wildlife isolates, but not for the other compartments (ANOVA: F = 4.940, df = 2, *p* = 0.017, Tukey post-hoc test: human-cattle: *p* = 0.261, human-wildlife: *p* = 0.013, cattle-wildlife: *p* = 0.299). The genetic differentiation of isolates among humans, cattle, and wildlife was significant (overall: FST = 0.071 ± 0.012, *p* < 0.00010; pairwise FST: human-cattle = 0.060, human-wildlife = 0.144, cattle-wildlife = 0.074, *p* = 0.017 for all). 

Then, the genetic diversity and differentiation in the three compartments were analyzed by including only isolates from patients with a “recent and local” risk of contamination. Therefore, all isolates from >50-year-old patients, from patients with a childhood contact risk, and from patients who may have been infected abroad were excluded. In this small human dataset (N = 17), genetic diversity was 0.678 ± 0.254, similar to the diversity of the whole human dataset. The genetic diversity difference was still significant between humans and wildlife (ANOVA: F = 3.907, df = 2, *p* = 0.036, Tukey post-hoc test: human-cattle: *p* = 0.360, human-wildlife: *p* = 0.028, cattle-wildlife: *p* = 0.360). Genetic differentiation also remained significant overall between human and wildlife isolates, and between cattle and wildlife (overall: FST = 0.067 ± 0.016, *p* < 0.0001; pairwise FST: human-cattle = 0.022, *p* = 0.033, human-wildlife = 0.135, *p* = 0.017, cattle-wildlife = 0.074, *p* = 0.017)

The DAPC (Figure 4a) confirmed that human isolates (blue) were more genetically diverse, based on the MIRU-VNTR profiles, than cattle (red) and wildlife (green) isolates. Animal isolates (cattle and wildlife) tended to group together. Human isolates were more disseminated and partially overlapped with animal isolates. When only isolates from <50-year-old patients who never travelled abroad were retained, they were grouped with animal isolates (Figure 4b).

#### 3.5.2. Analysis of the Genetic Diversity and Differentiation within the Three Main Spoligotypes Showed That:

-Within spoligotype SB0120, genetic diversity was 0.606 ± 0.173 in human (N = 45), 0.386 ± 0.191 in cattle (N = 357), and 0.196 ± 0.210 in wildlife (N = 48) isolates. Genetic diversity was significant between human and wildlife isolates (ANOVA: F = 9.091, df = 2, *p* = 0.001, Tukey post-hoc test: human-cattle: *p* = 0.080, human-wildlife: *p* = 0.001, cattle-wildlife: *p* = 0.144). Genetic differentiation also was significant among isolates of three host types (overall FST = 0.176 ± 0.020; pairwise FST: human-cattle = 0.199; human-wildlife = 0.308; cattle-wildlife = 0.130; *p* = 0.017 for all three);-Within spoligotype SB0121, genetic diversity was 0.557 ± 0.274 in human (N = 29), 0.444 ± 0.214 in cattle (N = 62), and 0.229 ± 0.214 in wildlife (N = 5) isolates. Genetic diversity was significant between human and wildlife isolates (ANOVA: F = 4.032, df = 2, *p* = 0.034, Tukey post-hoc test: human-cattle: *p* = 0.314, human-wildlife: *p* = 0.026, cattle-wildlife: *p* = 0.364). Genetic differentiation was overall significant, but pairwise tests could only be performed between human and cattle isolates due to the small wildlife sample size (overall: FST = 0.133 ± 0.034, *p* < 0.0001; pairwise FST: human-cattle = 0.132, *p* = 0.017; human-wildlife = 0.176, cattle-wildlife = 0.118, no *p* value computed);-Within spoligotype SB0134, genetic diversity was 0.594 ± 0.273 in human (N = 20), 0.326 ± 0.199 in cattle (N = 126), and 0.075 ± 0.004 in wildlife (N = 40) isolates. Genetic diversity was significant between humans and wildlife (ANOVA: F = 13.915, df = 2, *p* = 0.001, Tukey post-hoc test: human-cattle: *p* = 0.033, human-wildlife: *p* = 0.001, cattle-wildlife: *p* = 0.047). Genetic differentiation was overall significant and also for all host-type pairs (overall FST = 0.205 ± 0.030, *p* < 0.0001; pairwise FST: human-cattle = 0.257; human-wildlife = 0.49; cattle-wildlife = 0.081, *p* = 0.017 for all three).

### 3.6. Spatiotemporal Variation of the Population Structure (MIRU-VNTR Data)

Genetic differentiation was significant among years in all three hosts (humans: FST = 0.038 ± 0.100, *p* < 0.0001; cattle: FST = 0.039 ± 0.005, *p* < 0.0001; wildlife: FST = 0.179 ± 0.028, *p* = 0.0003). The genetic structure of the isolates was significantly different among regions in the cattle and wildlife compartments (FST = 0.200 ± 0.018, *p* < 0.0001; FST = 0.350 ± 0.069, *p* < 0.0001), but not in the human compartment (FST = 0.007 ± 0.005, *p* = 0.074). 

In the seven regions with N > 9 human and cattle isolates (see the list of regions, Figure 1), the differentiation between host compartments was more important than the geographical differentiation, although both were significant (host compartment: 10%, region level: 1% of the genetic variance; differentiation among host types within regions: FST= 0.105, *p* = 0.001; differentiation among regions: FST = 0.006, *p* = 0.007). Conversely, in regions with N > 17 cattle and wildlife isolates, genetic differentiation was higher among regions than among hosts (host compartment level: 1%, region level: 18% of genetic variance; differentiation among host types within regions: FST = 0.011, *p* = 0.005; differentiation among regions: FST = 0.178, *p* = 0.001).

### 3.7. Factors That Influence the Genetic Diversity and Differentiation in the Human Compartment

#### 3.7.1. Impact of the Country of Birth and Trips abroad 

Spoligotypes and information about the patient’s country of birth and travels abroad were known for 199 human isolates (102 patients were born in France and never travelled abroad, and 97 were born abroad and/or travelled abroad before the infection). Morocco (67%) and Algeria (18%) were the most common foreign countries associated with human cases. In isolates from patients who were born/and or had travelled abroad, 33 different spoligotypes were identified, particularly SB0120 (21.6%), SB0121 (19.6%), SB0265 (9.3%), and SB0134 (8.2%). In this group of patients, all people infected with an isolate carrying the SB0265 spoligotype were born in or had travelled to Morocco. In isolates from patients who were born in France and who had never travelled abroad, 49 different spoligotypes were detected, particularly SB0120 (24.5%), SB0134 (8.8%), and SB0121 (7.8%). SB0265 was detected only in one isolate (1.0%). MIRU-VNTR typing of 189 human isolates identified 98 different MIRU-VNTR profiles in isolates from patients who came from/visited a foreign country (n = 90), and 89 profiles in isolates from patients who did not (n = 99). The same distribution was obtained by combining spoligotypes and MIRU-VNTR profiles. Genetic differentiation between the isolates from patients who were born or/and had travelled abroad and patients who did not was also significant (FST = 0.020 ± 0.012, *p* < 0.0001). Genetic diversity was 0.651 ± 0.185 in strains isolated from patients who were born/travelled abroad and 0.680 ± 0.165 in patients who did not (Welsh’s *t*-test: *t* = −0.332, df = 13.828, *p* = 0.745).

#### 3.7.2. Genetic Differentiation of the Isolates from Patients with Pulmonary and Extrapulmonary Infections

Spoligotypes and information on the zoonotic TB clinical form were known for 214 isolates (n = 105 extrapulmonary form and n = 109 pulmonary form). In isolates from patients with extrapulmonary infection, 38 different spoligotypes were detected, particularly SB0120 (24.8%), SB0134 (13.3%), SB0121(12.4%), and SB0265 (8.6%). In isolates from patients with pulmonary infection, 53 different spoligotypes were identified, particularly SB0120 (19.3%), SB0121 (14.7%), and SB0134 (7.3%). SB0265 accounted for 2.8% of isolates in this group.

Information about the clinical form was known for the 204 fully genotyped isolates (n = 99 extrapulmonary and n = 105 pulmonary infections). In these two groups, 95 different MIRU-VNTR profiles and complete genotypes (extrapulmonary infection) and 103 MIRU-VNTR profiles and complete genotypes (pulmonary infections) were obtained. Population differentiation between isolates from these two groups (pulmonary vs extrapulmonary infections) was significant (FST = 0.013 ± 0.008, *p* = 0.013). The genetic diversity was 0.702 ± 0.153 in isolates involved in pulmonary infections and 0.639 ± 0.184 in isolates implicated in pulmonary infections (Welsh’s *t*-test: *t* = −0.742, df = 13.542, *p* = 0.471). 

## 4. Discussion

Zoonoses represent the major part of emerging or reemerging diseases worldwide. Their epidemiology is particularly complex because of the implication of different host compartments that may complicate the identification of reservoirs and transmission routes. In this study, we analyzed microbiological and genetic data on *M. bovis* isolates from humans, cattle, and wildlife collected in France over the course of 10 years. This dataset allowed us to investigate the genetic population structure of *M. bovis* in France, according to the host compartments, space, and time.

### 4.1. Prevalence of M. caprae Species 

In our study, few isolates of *M. caprae* were isolated from humans and none from cattle or wildlife. These data confirm that this species remains rare in France whatever the host, contrary to other European countries, such as Germany, where this species seems to be emerging in humans and animals [37,38,39,40].

### 4.2. Interhost Genetic Structure of M. bovis

First, based on spoligotyping, MIRU-VNTR profiles, and their combination, we compared the genetic diversity of the isolates in the human, cattle, and wildlife compartments. Genetic diversity was always higher in human isolates than in animal isolates. The relative number of genotypes (ratio between the number of genotypes and the number of samples) was three to 12 times (depending on the genotyping method) higher in isolates from humans than from cattle and wild animals. Genetic diversity also was significantly higher in human than in wildlife isolates. 

Second, differences in the genetic profiles were observed in the three host compartments. SB0120, SB0121, and SB0134 were frequent in all three host types, but with variable frequencies. SB0265 was regularly detected in human isolates, but extremely rare or absent in animal isolates. This spoligotype was mostly associated with extrapulmonary TB forms, suggesting contamination by ingestion or a preferential extrapulmonary tropism of these *M. bovis* isolates. The complete genotypes (i.e., the combination of spoligotypes and MIRU-VNTR profiles) also differed in humans and animals. For instance, the most abundant profiles in cattle, and also represented in wild animals (SB0120-5-3-5-3-9-4-5-6, SB120-5-5-4-3-11-4-5-6, and SB0134-6-4-5-3-6-4-3-6), were absent in the human dataset. Moreover, isolates of the F4 family were rare, and were not detected in human isolates from the south of France, although many bovines in southern France carry F4 isolates. Additionally, MIRU-VNTR-based genetic differentiation was significant among the three host types. However, the differentiation between cattle and wildlife isolates could be mostly explained by the spatial variation in the sample collection. This might be due to the fact that the sampling of wildlife isolates was geographically biased (only in regions with high bovine TB prevalence in cattle) and limited (few isolates genotyped). In agreement, recent studies based on intensified wildlife screening and surveillance (in terms of number of isolates and number of geographic areas surveyed) since 2010 found an overlap of bovine TB outbreaks in wildlife and in cattle with a circulation of common *M. bovis* genotypes between these compartments. In the DAPC plots, human isolates overlapped only partly with animal isolates. All these data suggest that the genetic pool of human and animal isolates significantly differed, possibly because 49% of included patients might have contracted the disease abroad. Indeed, spoligotype SB0265 is frequently identified in isolates from cattle in Morocco [41], but has been described also in isolates from boars, deer, cattle, and goats in six regions of Portugal [42], in a boar isolate in southern Spain [43], and in a human and several bovine isolates in northern Spain [9]. This spoligotype was also detected in a study on slaughtered animals in Tunisia [44]. Epidemiological studies in North Africa are needed to identify the circulating spoligotypes that may spread to other regions, particularly in humans. Additionally, some patients may have experienced reactivations of infections contracted in their youth. These two situations match the epidemiological data available for Europe and France [9,13,14,45]. The observed genetic differentiation could also be due to the host specialization of isolates. Indeed, studies on strains involved in human-to-human transmission revealed the presence of specific adaptations in virulence genes, to compensate from the previous loss in infectivity towards humans [22,46]. This re-adaptation to the human host could have led to the genetic differentiation of some lineages among the host compartments in our dataset. However, our results do not support this hypothesis. First, several genotypes were shared by human and cattle isolates. Although we lack contact tracking information to confirm possible transmission between different compartments, these shared genotypes suggest an ongoing circulation of bovine TB between the human and the cattle compartments. Moreover, when considering only the isolates from patients with probable “recent and local” transmission, we observed that the genetic differentiation between human and cattle isolates was no longer significant. However, we cannot formally rule out the possibility that some of the studied isolates were re-adapted to the human host. We should study the virulence factors, particularly in isolates involved in human-to-human transmission, to determine whether some of our human isolates were re-adapted.

### 4.3. Spatiotemporal Changes of M. bovis in the Three Host Compartments

Our dataset was characterized by an heterogenous number of isolates from the different French regions, and by a difference in distribution of human and cattle isolates. The variation was also important for wildlife isolates, but this was due to the fact that before 2005, wildlife sampling was performed only in areas with high *M. bovis* prevalence in cattle [8]. Many human isolates were from Île de France, the region of Paris (the capital city), and Rhône-Alpes, the region of Lyon (the third biggest French city). Conversely, most cattle isolates were from the south of France and Bourgogne. This spatial difference between human and cattle isolates is in line with the fact that most human cases in France do not seem to be directly caused by a recent contamination on the French territory. However, we did not have any information on the patients’ movements within France. Therefore, human isolates could have been collected in an urbanized region where patients were treated, although they had been contaminated in another region. Indeed, in our dataset, eight isolates from patients were genetically identical to isolates from cattle collected in other regions. This may be due to the fact that these isolates might have been circulating also in cattle in the region where the patient was infected and treated, but they were not represented in our sample, or that patients were contaminated while visiting another region. An isolate could also have been circulating in a region in the past, although still represented in other areas. Genotypic diversity was spatially variable in all host compartments. In human and wildlife samples, genotypic diversity hotspots correlated with regions where the highest number of isolates were collected. This was not always true for cattle isolates, for which the highest diversity was mostly observed in samples from southern France and Bourgogne, unlike in human isolates. Interestingly, geographic differentiation was significant within cattle and wildlife isolates, but not within human isolates, possibly because human *M. bovis* epidemiology is mostly influenced by reactivated or imported strains, with minimal links with the local *M. bovis* epidemiology in cattle.

Lastly, temporal variations were observed in all three host compartments. In the human and wildlife compartments, temporal variations of genotypic diversity were linked to the sampling size. Conversely, in the cattle compartment, we observed a global decrease in diversity despite a stable sample size. This suggests that control measures efficiently reduce the number of clones circulating in cattle (see also Hauer et al. 2015) [8]. Moreover, genetic differentiation significantly varied among years. In human isolates, genetic differentiation could be due to random reactivation or variations in the imported strains. In cattle isolates, differentiation could be explained by genetic drift, driven by the gradual removal of clones by control measures or changes in farming practices over time [8]. In wildlife isolates, differentiation could be a consequence of the genetic drift observed in cattle, because the reduction of genetic diversity in cattle strains may lead to a decrease in the diversity of strains that spillover to wild animals. However, it could have also been influenced by the sampling bias. 

For many years the combination of spoligotyping + MIRU-VNTR has been considered as the reference method for the realization of molecular epidemiological studies of *M. bovis* [47,48,49,50,51]. Recently, whole-genome sequencing (WGS) has become the preferential technique to inform outbreak response through contact tracing and source identification for many infectious diseases, including bovine tuberculosis [52]. In their review, Guimaraes et al. describe *M. bovis* genotyping techniques and discuss current standards and challenges in the use of *M. bovis* WGS for transmission investigation, surveillance, and global lineage distributions [52]. In our study, WGS could be provided as the solution to finely resolve transmission patterns happening at the individual herd level, in clusters of small spatial extent, particularly in countries, such as France, where bTB prevalence is almost null, and re-introduction outbreaks occur due to a single-sourced *M. bovis* strain [52,53]. Because of the high resolving power, WGS could have, in our study, differentiated *M. bovis* isolates with identical spoligotyping + MIRU-VNTR genotypes.

In conclusion, our study suggests that in France, *M. bovis* epidemiology in humans is mostly driven by past (reactivations) and foreign (contaminations abroad) dynamics, but that exchanges between the human and cattle compartments remain possible. Spatiotemporal variations of the genetic structure were observed in cattle and wildlife isolates (see also Hauer et al. 2015) [8], whereas human isolates varied in time but not in space. Our data suggest that *M. bovis* genetic composition, spatial, and temporal dynamics are different in humans and animals in France. Prevention methods have drastically reduced the risks of human infection in France, but they do not prevent people from becoming infected abroad. Our study highlights the importance of considering *M. bovis* control as a global issue and to take into account also human population movements worldwide. In France, a large number of *M. bovis* infections in humans concerns patients who were born in or who regularly visited endemic countries (mainly North Africa). It would be interesting to study the genotypes circulating in humans and animals in these regions to better understand *M. bovis* sources and the modes of transmission to humans. Moreover, our sampling might have underestimated the wildlife role and sampling efforts in this compartment must be increased. Despite the human/animal genetic differentiation of the isolates, our study does not provide a strong support to the hypothesis of *M. bovis* strains specifically adapted to different host compartments. More analyses on virulence factors are needed to test this hypothesis more extensively. 

Maps created by adapting the blank maps provided by http://blog.comersis.com/articles/office-map, accessed on 1 March 2022.

## Figures and Tables

**Figure 1 pathogens-12-00548-f001:**
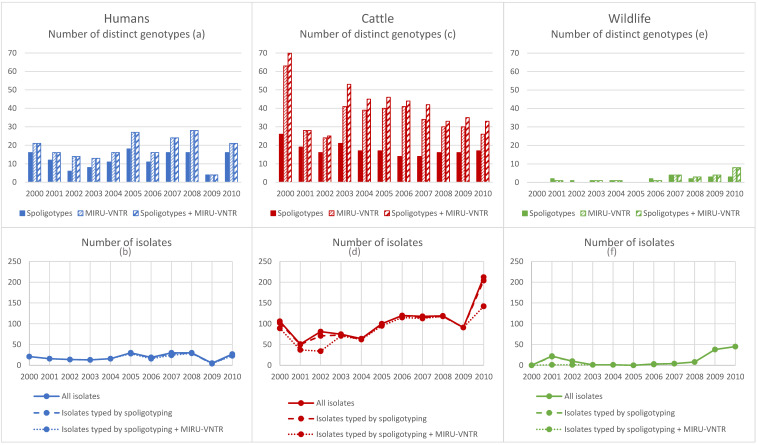
Temporal variations in the genotypic richness and in the number of M. bovis and M. caprae (only in human) isolates in the three host compartments (Human (**a**,**b**); Cattle (**c**,**d**); Wildlife (**e**,**f**)).

**Figure 2 pathogens-12-00548-f002:**
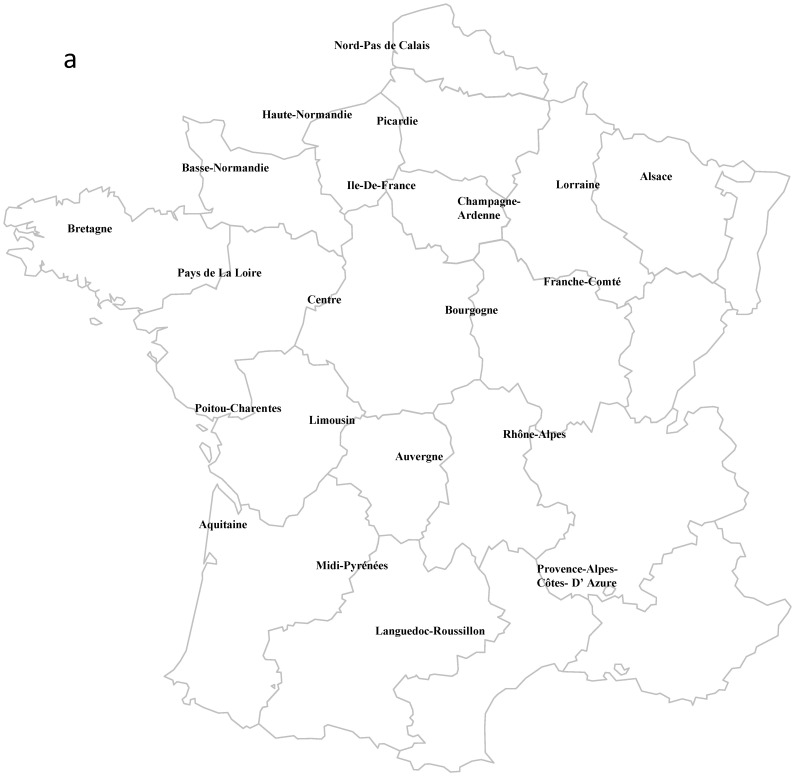
(**a**) Spatial variations in the genotypic richness and in the number of *M. bovis* and *M. caprae* (only in human) isolates in the three host compartments (Human (**b1**); Cattle (**b2**); Wildlife (**b3**)).

**Figure 3 pathogens-12-00548-f003:**
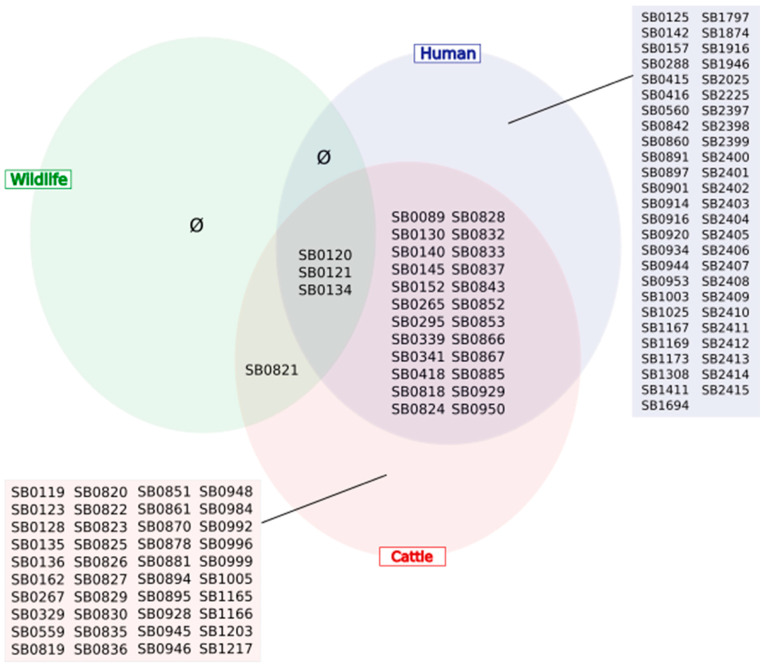
Distribution of spoligotypes represented at least once in the three host compartments (human-cattle-wildlife).

**Figure 4 pathogens-12-00548-f004:**
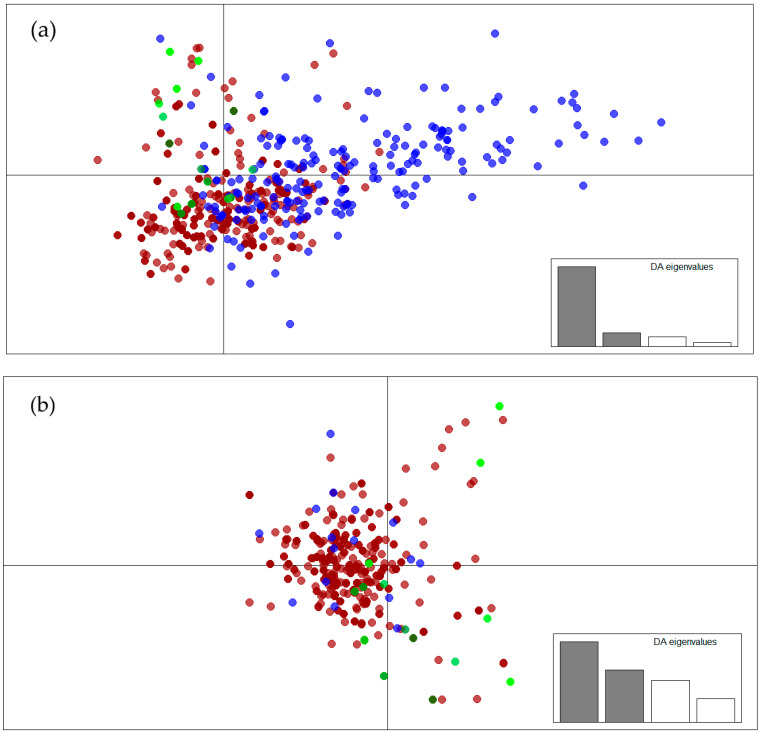
Discriminant analysis of the principal components showing all genotyped isolates (**a**) from the three host compartments (blue, human isolates; red, cattle isolates; green, wildlife isolates), and all genotyped isolates (**b**) for cattle (red) and wildlife (green,) but only isolates from patients with *M. bovis* TB possibly acquired recently in France (blue).

## Data Availability

Not applicable.

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
