# Peer review of "Genetic Diversity and Population Structure of Mycobacterium bovis at the Human-Animal-Ecosystem Interface in France: “A One Health Approach”"

_pathogens, 2023, doi:10.3390/pathogens12040548_

Round 1
Reviewer 1 Report
This manuscript analyzes the genetic diversity of M. bovis isolates detected in humans, cattle and wildlife between 2000-2010 in France, trying to identify the interspecies transmission and spatiotemporal relationships of these isolates, which represent an important zoonotic disease.
The study is clear and the results will be of interest to researchers and health institutions belonging to the human, animal and environmental fields.
I have only minor comments
- All scientific names should be in italics
- References should be cited correctly, because in the current format it seems confusing
- The top graphs in Figure 1 require legends.
- The discriminatory power of the spoligotyping and MIRU-VNTR techniques is different, so in light of the results, the authors should say something about their usefulness, pros and cons for genotyping M. bovis isolates in the compartments that have been analyzed.
Author Response
Reviewer #1
This manuscript analyzes the genetic diversity of M. bovis isolates detected in humans, cattle and wildlife between 2000-2010 in France, trying to identify the interspecies transmission and spatiotemporal relationships of these isolates, which represent an important zoonotic disease.
The study is clear and the results will be of interest to researchers and health institutions belonging to the human, animal and environmental fields.
- Minor comments
Comment N°1: “All scientific names should be in italics”
Response N°1: Correction done.
Comment N°2: References should be cited correctly, because in the current format it seems confusing
Response N°2: Correction done.
Comment N°3: The top graphs in Figure 1 require legends.
Response N°3: Correction done. the legend for Figure 1 now reads as follows:
" Figure 1: Temporal variations in genotypic richness and in the number of M. bovis isolates in the three host compartments (Human 1a, 1b; Cattle 2a, 2b; Wildlife 3a, 3b). Figure 1 is attached. The figure with these corrections has been attached
Comment N°4: The discriminatory power of the spoligotyping and MIRU-VNTR techniques is different, so in light of the results, the authors should say something about their usefulness, pros and cons for genotyping M. bovis isolates in the compartments that have been analyzed.
Response N°4: A paragraph on this topic has been added in the introduction to the line 123. “The choice of the optimal molecular markers in accordance with the scope of the study also depends on the space and time scales in which the data were collected or explored. In fact, the speed of evolution (molecular clock) of a given marker, which conditions its power of resolution. Nevertheless, the resolution power of each marker is a function of the organism and the species under study. For M.bovis, from a resolution power point of view, MIRU-VNTR appeared very appropriate for short-term epidemiology studies, whereas spoligotyping is more suitable for long-term epidemiology studies. Nevertheless, several studies showed clearly that using multiple methods for molecular epidemiology is necessary [25-28]. Several authors recommend spoligotyping associated with MIRU-VNTR for molecular epidemiology studies [25-28].
Reviewer 2 Report
Brief summary:
The manuscript deals with the challenges of understanding TB epidemiology.
The authors have carried out a study to assess and compare the genetic diversity of M. bovis isolates obtained from infected humans and animals in France.
They compared M. bovis genetic diversity in isolates collected from infected humans, cattle and wildlife by spoligotyping and MIRU-VNTR typing, two standard genetic fingerprinting and population genetic methods. They assessed M. bovis genetic structure within and among the different host groups, and across time and space.
The results of the study show that most of the genotypes detected in the human isolates were absent in the cattle and wild animal isolates, probably because in the patients the M. bovis infection was contracted abroad or was the reactivation of an old lesion. Therefore, they did not correspond to the genetics present in France during the study period.
However, some human-cattle interactions did occur because some genotypes were common to both compartments.
The study provides some new elements to improve understanding of the epidemiology of M. bovis in France.
Broad comments:
The reading of the manuscript is quite fluent and pleasant, I just have to point out some formatting problems:
- The missing use of italics for the names of mycobacteria e.g. Mycobacterium bovis
- The absence of brackets in the text for bibliographical references;
- It is often difficult to see whether there are headings and/or sub-headings, I point out later the lines where this happens.
All the figures should be improved except for figure 3.
It might be useful to make some tables to better describe the various genotypes in the three group analyzed (human, cattle, wildlife).
The following two articles (to be cited in the paper) could be referred to:
Hauer, A., Michelet, L., De Cruz, K., Cochard, T., Branger, M., Karoui, C., et al. (2016). MIRU-VNTR allelic variability depends on Mycobacterium bovis clonal group identity. Infect. Genet. Evol. 45, 165–169. doi: 10.1016/j.meegid.2016. 08.038
Boniotti, M. B., Goria, M., Loda, D., Garrone, A., Benedetto, A., Mondo, A., et al. (2009). Molecular typing of Mycobacterium bovis strains isolated in Italy from 2000 to 2006 and evaluation of variable-number tandem repeats for geographically optimized genotyping. J. Clin. Microbiol. 47, 636–644. doi: 10.1128/JCM.01192-08
Specific comments:
Abstract
Line 41: place replace “controlled” with “reduced”
Introduction
Mycobacterium caprae should be briefly mentioned, and some historical references that report M.caprae in humans should be cited, e.g. Prodinger, 2002 and/or Kubica, 2003.
Lines 78-82: I think that you are not only talking about Spain but also about England and Ireland (bibliographic reference 12)
Line 89: place remove “is present and” or improve the sentence.
Material and methods
Lines 226-230: Perhaps it would be better to head and keep the wildlife separate, as "Human dataset" and "Animal dataset" make a “Wildlife dataset".
Line 260: Please add a reference
Line 263-264 “Spoligotype frequencies and comparisons of complete genotypes (spoligotypes + MIRU-VNTRs)”: is this a title? Or a subtitle?
Figure 1 can be shortened
Figures 2, 3 and 4 must be set up differently, e.g. working on a single map, using spheres or balls of different colours and sizes (not bars).
Results
Line 370-371 “Diversity of MIRU-VNTR profiles and combination of spoligotypes and MIRU- 370 VNTR profiles”: is this a title? Or a subtitle?
Figure 4 is unclear, the legend needs to be improved with a box describing the colour and the species.
Line 515-516-517 “Factors that influence genetic diversity and differentiation in the human compartment -Impact of the country of birth and trips abroad”: are this titles? Or subtitles?
Idem for lines 537-538
Supplementary Materials: in Table S1 human data 'Cluster' written without 's'
Reference
If possible, some more recent bibliographical references should be added (the most recent articles date back to 2018)
Check well the reported references in accordance with the format required by “Pathogens-MDPI” Journal.

Author Response
Reviewer #2
Brief summary: The manuscript deals with the challenges of understanding TB epidemiology. The authors have carried out a study to assess and compare the genetic diversity of M. bovisisolates obtained from infected humans and animals in France. They compared M. bovis genetic diversity in isolates collected from infected humans, cattle and wildlife by spoligotyping and MIRU-VNTR typing, two standard genetic fingerprinting and population genetic methods. They assessed M. bovis genetic structure within and among the different host groups, and across time and space.
The results of the study show that most of the genotypes detected in the human isolates were absent in the cattle and wild animal isolates, probably because in the patients the M. bovisinfection was contracted abroad or was the reactivation of an old lesion. Therefore, they did not correspond to the genetics present in France during the study period. However, some human-cattle interactions did occur because some genotypes were common to both compartments. The study provides some new elements to improve understanding of the epidemiology of M. bovis in France.
- Broad comments:
The reading of the manuscript is quite fluent and pleasant, I just have to point out some formatting problems:
Comment N°1:The missing use of italics for the names of mycobacteria e.g. Mycobacterium bovis
Response N°1: Correction done.
Comment N°2:The absence of brackets in the text for bibliographical references.
Response N°2: Correction done.
Comment N°3: It is often difficult to see whether there are headings and/or sub-headings, I point out later the lines where this happens.
Response N°3: The titles and subtitles have been numbered for a more fluid reading
Comment N°4: All the figures should be improved except for figure 3.
Response N°4: We have tried to be more precise about the legends of the different figures.
Comment N°5: It might be useful to make some tables to better describe the various genotypes in the three group analyzed (human, cattle, wildlife).
Response N°5: To facilitate the reading of the article, all the information concerning the genotypes in each compartment is described in the tables of the supplementary material: TableS1-Human-Informations; TableS2-Cattle-Informations; TableS1-Wildlife-Informations.
Comment N°6: The following two articles (to be cited in the paper) could be referred to:
Hauer, A., Michelet, L., De Cruz, K., Cochard, T., Branger, M., Karoui, C., et al. (2016). MIRU-VNTR allelic variability depends on Mycobacterium bovis clonal group identity. Infect. Genet. Evol. 45, 165–169. doi: 10.1016/j.meegid.2016. 08.038
Boniotti, M. B., Goria, M., Loda, D., Garrone, A., Benedetto, A., Mondo, A., et al. (2009). Molecular typing of Mycobacterium bovis strains isolated in Italy from 2000 to 2006 and evaluation of variable-number tandem repeats for geographically optimized genotyping. J. Clin. Microbiol. 47, 636–644. doi: 10.1128/JCM.01192-08
Response N°6: These 2 references have been added to the texts line 113, 120 and 121 as well as in the bibliography section.
- Specific comments:
- Abstract
Comment N°7: Line 41: place replace “controlled” with “reduced”
Response N°7: Correction done.
- Introduction
Comment N°8: Mycobacterium caprae should be briefly mentioned, and some historical references that report M.caprae in humans should be cited, e.g. Prodinger, 2002 and/or Kubica, 2003.
Response N°8: The species M.caprae has been mentioned and discussed on line 575. The references of Prodiger et al. and Kubica et al. have been added (ref 39 and 40).
Comment N°9: Lines 78-82: I think that you are not only talking about Spain but also about England and Ireland (bibliographic reference 12)
Response N°9: We agree with the reviewer. The sentence line 82 has therefore been changed: " However, several genetic fingerprinting studies in Spain, England Ireland challenged this assumption by showing that a important number of genotype of human isolates was identical or similar to that of bovine isolates”.
Comment N°10: Line 89: place remove “is present and” or improve the sentence.
Response N°10: Line 89 : “is present and" has been removed from the sentence.
- Material and methods
Comment N°11: Lines 226-230: Perhaps it would be better to head and keep the wildlife separate, as "Human dataset" and "Animal dataset" make a “Wildlife dataset".
Response N°11: done line 232.
Comment N°12: Line 260: Please add a reference
Response N°12: The reference is present. “Team, R. C., R: A Language and Environment for Statistical Computing. Vienna Austria R Found. Stat. Comput. Available online at https://www.R-project.org/. 2016. »
Comment N°13: Line 263-264 “Spoligotype frequencies and comparisons of complete genotypes (spoligotypes + MIRU-VNTRs)”: is this a title? Or a subtitle?
Response N°13: The sentence “Spoligotype frequencies and comparisons of complete genotypes (spoligotypes + MIRU-VNTRs)” has been deleted from the sentence
Comment N°14: Figure 1 can be shortened
Response N°14: The figure was not shortened, but clarifications were made by becoming Figure 1a, 1b, 2a, 2b, and 3c, 3c.
Comment N°15: Figures 2, 3 and 4 must be set up differently, e.g. working on a single map, using spheres or balls of different colours and sizes (not bars).
Response N°15: This change seems a bit complicated to us and the figure seems quite understandable as it is.
-Results
Comment N°16: Line 370-371 “Diversity of MIRU-VNTR profiles and combination of spoligotypes and MIRU- 370 VNTR profiles”: is this a title? Or a subtitle?
Response N°16: This is indeed title 3.3. The correction has been done.
Comment N°17: Figure 4 is unclear, the legend needs to be improved with a box describing the colour and the species.
Response N°17: The colors corresponding to the hosts (human, cattle and wildlife) are already specified in the legend.
Comment N°18: Line 515-516-517 “Factors that influence genetic diversity and differentiation in the human compartment -Impact of the country of birth and trips abroad”: are this titles? Or subtitles?
Response N°18: It consists of one title and 2 subtitles (“3.7. Factors that influence genetic diversity and differentiation in the human compartment; 3.7.1.Impact of the country of birth and trips abroad; 3.7.2.Genetic differentiation of isolates from patients with pulmonary and extrapulmonary infection.”
Comment N°19: Idem for lines 537-538
Response N°19: It consists of one title and 2 subtitles. (“3.7. Factors that influence genetic diversity and differentiation in the human compartment; 3.7.1.Impact of the country of birth and trips abroad; 3.7.2.Genetic differentiation of isolates from patients with pulmonary and extrapulmonary infection.”
Comment N°20: Supplementary Materials: in Table S1 human data 'Cluster' written without 's'
Response N°20: done.
- Reference
Comment N°21: If possible, some more recent bibliographical references should be added (the most recent articles date back to 2018)
Response N°21: More recent references have been added (reference numbers: 47 to 53).
Comment N°22: Check well the reported references in accordance with the format required by “Pathogens-MDPI” Journal.
Response N°22: Done.
Reviewer 3 Report
The manuscript describes a thorough, comprehensive analysis of the genetic diversity and structure of Mycobacterium bovis, the primary cause of animal tuberculosis (TB) and also responsible for a small proportion of cases of tuberculosis in humans (zoonotic TB) in developed countries. This collaborative study represents an important contribution to the molecular epidemiology of Mycobacterium bovis infections in human and animals and to our understanding of epidemiology of zoonotic TB in France, a country that although recognised as officially free from bovine TB still contains localised pockets of TB in cattle and wildlife reservoirs.
The manuscript is well written, the methods seem sound and are well described, and the authors acknowledge the main limitations of their study in the Discussion. Therefore, I recommend acceptance for publication in this journal, subject to minor revisions.
I would ask that the authors include a brief reference to whole-genome sequencing (WGS) of M. bovis isolates in the Discussion and how this technique, if applied to French M. bovis isolates, could have enhanced the results and conclusions of their analysis. WGS is rapidly becoming the standard method for analysis of genetic relatedness between M. bovis isolates from animals and humans and it provides greater resolution and discrimination than spoligotyping and MIRU-VNTR typing.
My more detailed comments, which are mostly minor text edits and stylistic suggestions, are shown as annotations (yellow text notes) inserted in the PDF copy of the manuscript supplied to me by the editors. The number at the beginning of each note indicates the line of text in the PDF that the comment refers to.

Author Response
Reviewer #3
The manuscript describes a thorough, comprehensive analysis of the genetic diversity and structure of Mycobacterium bovis, the primary cause of animal tuberculosis (TB) and also responsible for a small proportion of cases of tuberculosis in humans (zoonotic TB) in developed countries. This collaborative study represents an important contribution to the molecular epidemiology of Mycobacterium bovis infections in human and animals and to our understanding of epidemiology of zoonotic TB in France, a country that although recognised as officially free from bovine TB still contains localised pockets of TB in cattle and wildlife reservoirs.
The manuscript is well written, the methods seem sound and are well described, and the authors acknowledge the main limitations of their study in the Discussion. Therefore, I recommend acceptance for publication in this journal, subject to minor revisions.
Comment N°1: I would ask that the authors include a brief reference to whole-genome sequencing (WGS) of M. bovis isolates in the Discussion and how this technique, if applied to French M. bovis isolates, could have enhanced the results and conclusions of their analysis. WGS is rapidly becoming the standard method for analysis of genetic relatedness between M. bovis isolates from animals and humans and it provides greater resolution and discrimination than spoligotyping and MIRU-VNTR typing.
Response N°1: As suggested by the reviewer, a paragraph with references corresponding to this topic has been added in the discussion line 676: " For many years the combination of spoligityping + MIRU-VNTR has been con-sidered as the reference method for the realization of molecular epidemiological studies of M.bovis [47-51]. Recently, whole-genome sequencing (WGS) has become the preferential technique to inform outbreak response through contact tracing and source identification for many infectious diseases including bovine tuberculosis [52]. In their review of Guimaraes et al. describe M. bovis genotyping techniques and discuss current standards and chal-lenges of the use of M. bovis WGS for transmission investigation, surveillance, and global lineages distribution [52]. In our study, WGS could be provided there solution to finely resolve transmission patterns happening at the individual herd level, in clusters of small spatial extent, in particularly in country as France where bTB prevalence is almost null and re-introduction outbreaks occur due to a single-sourced M. bovis strain [52, 53]. many studies show that WGS is useful to differentiate M. tuberculosis strains with identical MIRU-VNTR genotypes, proving superior resolution. Because of the high resolving power, WGS could have allowed in our study to differentiate M. bovis isolates with identical spoligotyping + MIRU-VNTR genotypes.”
My more detailed comments, which are mostly minor text edits and stylistic suggestions, are shown as annotations (yellow text notes) inserted in the PDF copy of the manuscript supplied to me by the editors. The number at the beginning of each note indicates the line of text in the PDF that the comment refers to.
Comment N°1: Line 90. "between" instead of "among"
Response N°1: done.
Comment N°2: 91. Do you mean "infected through contact with human excreta contaminated with M. tuberculosis" (i.e. an antropozoonosis)?
Response N°2: Effect the sentence has been changed to clarify the meaning: “Moreover, cases of domestic animals contaminated with M.tuberculosis by human (i.e. an anthropozoonosis) excreta have been reported [20].”
Comment N°3: 92. Replace "wild hosts" with "wildlife reservoirs of M. bovis"
Response N°3: done.
Comment N°4: 96. replace "specialized" with "adapted" or "could show more affiinity"
Response N°4: done.
Comment N°5: 103, 116, 145, 320, 342. Use "wild boar" (singular, not plural, as with "deer".)
Response N°5: done.
Comment N°6: No mention of M. caprae in the captions to Figures 1 and 2. Should this not be added there?
Response N°6:
Comment N°7: 177. Please correct the numbering of the three figures in the Figure 1 caption, so that it reads "Human 1a, Cattle 1b; Wildlife 1c" and matches the numbers in the figure legends. Are there any data missing from the upper graph in figure 1c (i.e. number of distinct M. bovis genotypes in wildlife)? The bar chart is blank. If no data are missing, explain the absence of bars in the Figure 1 caption. What do the different patterns of the bars (i.e. solid, cross-hatching, bold cross-hatching) denote?
Response N°7: Correction done. The legend for Figure 1 now reads as follows:
" Figure 1: Temporal variations in genotypic richness and in the number of M. bovis isolates in the three host compartments (Human 1a, 1b; Cattle 2a, 2b; Wildlife 3a, 3b). Figure 1 is attached. The figure with these corrections has been attached. There is no missing data in Figure 1c. It is during the layout done by the journal that the data disappeared.
Comment N°8: 200. Correct the numbering of the three figures in the Figure 2 caption, so that it reads "Human 2a, Cattle 2b; Wildlife 2c" and matches the numbers in the figure legends.
Response N°8: done.
Comment N°9: 213: After "2016", add a reference to the large map in Figure 2,
Response N°9: done.
Comment N°10: 214: "travel to" instead of "travels in".
Response N°10: done.
Comment N°11: 215: Add "of TB" after "form".
Response N°11: done.
Comment N°12: 225. Delete "previous". Insert the words: "test or" between "skin" and "interferon".
Response N°12: done.
Comment N°13: 226. Replace "slaughter" with "post-mortem meat"
Response N°13: done.
Comment N°14: 228. Delete "(see Hauer et al. 2015)". Leaving the number of the relevant reference at the end of the sentence ("8") is sufficient.
Response N°14: done.
Comment N°15: 251. Why two isolates? Do you mean ONE isolate for each spoligotype profile detected from the same herd?
Response N°15: yes.
Comment N°16: 262. The previous section (Genotyping methods) was 2.4. Should this not be numbered 2.5, with the subsequent section in the M&M (MIRU-VNTR-based genetic differentiation analyses) renumbered 2.6?
Response N°16: the correction was done.
Comment N°17: 263. Is this an incomplete sentence, or an alternative title for section 2.4?
Response N°17: It is a subtitle, the correction was done.
Comment N°18: 285. Replace "in function of" with "according to", or simply "by", i.e. "by sampling year and by region".
Response N°18: done.
Comment N°19: 318. Delete "the whole", or replace those words with "metropolitan".
Response N°19: done.
Comment N°20: 330. Replace "livestock" with "cattle" or "bovine", for consistency with the terminology used elsewhere in the manuscript and to avoid confusion (all the livestock samples belonged to cattle).
Response N°20: done.
Comment N°21: 370-371. Is this another subsection title? If so, it needs to be numbered as 3.2.3.
Response N°21: done.
Comment N°22: 447. Replace "contaminated" with "infected".
Response N°22: done.
Comment N°23: The two graphs need to be labelled "4a" and "4b" respectively.
Response N°23: On the document provided to the journal the 2 graphs are annotated 4a and 4b.
Comment N°24: Should this not be the title of a new sub-section (3.5)?
Response N°24: done.
Comment N°25: 599. Replace "a disease" with "M. bovis infection".
Response N°25: done.
Comment N°26: Replace "contamination" with "infection".
Response N°26: done.
Comment N°27: 656 and 662. Replace "contaminations" with "infection".
Response N°27: done.
Comment N°28: The format of ref. 14 should match t hat of ref. 13.
Response N°28: done.